# UFO: Chain-of-Evaluation for Omni-Condition Alignment in Multi-Modal Image Generation

**Danning Zhang** [* 1]  **Yijing Lin** [* 1]  **Shuhan Zhuang** [* 1]
**Mengqi Huang** [✉ 1]  **Shaojin Wu** [2]  **Shancheng Fang** [3]  **Zhendong Mao** [1 4]

## Abstract

Multi-modal image generation, particularly subject-driven customization, has garnered growing attention in recent years. Despite the rapid advancement of generative models, their evaluation remains largely lagging. Existing methods, whether embedding-based or Multi-modal Large Language Model (MLLM)-based, evaluate alignment with each modal condition in isolation, which contradicts the simultaneous condition alignment objective of multi-modal image generation, leading to poor consistency with human judgments. To address this challenge, we propose **UFO**, the first **UniF**ied framework for **O**mni-condition alignment simultaneous evaluation. Specifically, UFO introduces a novel Atomized Chain-of-Evaluation paradigm, *i.e.*, it first decomposes omni-condition alignment into a sequential chain of fine-grained, disentangled Atomic Evaluation Units (AEUs), categorizes them into distinct modality-relevance classes, and then employs general or dedicated functional calls for accurate verification of different AEU types. Experimental results demonstrate that UFO achieves the highest correlation with human evaluation preferences, delivering an average improvement of 15.25%. Furthermore, we present UFO-Bench, a dedicated benchmark designed to holistically evaluate the performance of existing customization models under the diverse mutual interactions of textual and visual conditions. **Project Page:** https://github.com/UFOTeamwork/UFO.git

---

[*]Equal contribution [1] University of Science and Technology of China, Hefei, China [2] ByteDance Inc., Beijing, China [3] Shenzhen University, Shenzhen, China [4] Institute of Artificial Intelligence, Hefei Comprehensive National Science Center, Hefei, China . Project Lead: Shaojin Wu. Correspondence to: Mengqi Huang <huangmq@ustc.edu.cn>.

*Proceedings of the 43ʳᵈ International Conference on Machine Learning*, Seoul, South Korea. PMLR 306, 2026. Copyright 2026 by the author(s).

## 1. Introduction

Multi-modal image generation (DeepMind, 2025; Mao et al., 2025a; ByteDance, 2025; Wu et al., 2025c;a) has achieved significant progress in recent years, driven by advances in large-scale diffusion models (Peebles & Xie, 2023; Labs, 2024). These models are capable of generating high-quality visual content based on diverse cross-modal conditions, including textual conditions and visual conditions from reference images. The advancements have unlocked a plethora of creative applications, spanning personalized content generation, visual storytelling, *etc.* As generative models become increasingly expressive and controllable, reliable evaluation has emerged as a critical bottleneck for both research and real-world deployment.

Among various multi-modal image generation tasks, subject-driven customization (Huang et al., 2024; Mao et al., 2025b; Tan et al., 2025a;b) has drawn growing attention in recent years. It takes a text and images of a specific subject as input, aiming to generate new images that align with both *textual semantics* and the *subject's visual attributes that are not edited by the text*. In contrast to other text-image conditioned generation, such as image editing (Jiang et al., 2025; Qu et al., 2025), subject-driven customization requires re-generating the target visual subject in a **free-form manner**, *i.e.*, the generated subject may vary in pose, position, *etc.*, making it more challenging to conduct accurate evaluations. Specifically, such variations not only make it difficult to judge the subject's visual alignment, but also further complicate the assessment of its adherence to both textual and visual alignment simultaneously.

Existing evaluation for subject-driven customization can be categorized into two streams, *i.e.*, the traditional embedding-based evaluators (Salimans et al., 2016; Heusel et al., 2017) and Multi-modal Large Language Model (MLLM)-based evaluators (Peng et al., 2024; Deng et al., 2025b; Wang et al., 2025a). Traditional embedding-based evaluators, such as CLIP (Hessel et al., 2021) and DINO (Zhang et al., 2022), calculate the feature similarity between generated and reference images and assume that higher visual similarity implies better subject preservation. MLLM-based evaluators, such as VIEScore (Ku et al., 2024) and Dreambench++ (Peng

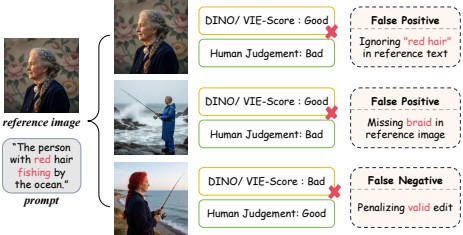

*Figure 1.* Limitations of isolated evaluation in subject-driven image generation. The top two rows show that existing evaluators fail to detect errors (ignoring the "red hair" prompt or missing the "braid"). The bottom row shows that evaluators penalize the successful edit that is correctly rated by humans.

et al., 2024), typically query the MLLM to assign a holistic score for evaluating textual and visual alignment. The common limitation of existing evaluators is that they all evaluate alignment with each modal condition *in isolation.*

In this study, we argue that the existing isolated evaluation paradigm is inherently *contractive to the simultaneous condition alignment objective* of multi-modal image generation, resulting in poor consistency with human evaluation, and thereby substantially restricts the advancement of subject-driven customization research. The reason for this is that subject-driven customization requires generated subjects to adhere to both textual and visual conditions, where textual conditions invariably involve modifications to specific attributes of the subject in the original reference image. Isolated evaluation, by contrast, only compares the visual appearance of subjects in reference and generated images, and thus ignores the intended modifications specified by textual conditions, ultimately leading to false positive and false negative issues. As shown in Figure 1, existing evaluators assign high scores to generated images that closely resemble the reference subject even when they ignore textual requirements (*e.g.*, failing to change hair color), resulting in false positives. Furthermore, they fail to capture fine-grained identity modifications, leading to other false positives. Conversely, valid generations that correctly apply the textual edit while preserving the subject's identity are often penalized due to reduced visual similarity, leading to false negatives.

To address this problem and establish a reliable, automatic, and reusable evaluation paradigm with benchmarks for effective measurement of the progress in subject-driven customization, we propose **UFO**, a **Uni**Fied framework for **O**mni-condition alignment simultaneous evaluation, presented here for the first time, along with **UFO-Bench**, which encompasses diverse multi-modal conditions' interaction scenarios to comprehensively assess the performance of existing customization models. Specifically, to tackle the challenges posed by the diverse potential interactions between distinct modality conditions (*i.e.*, textual and visual conditions), UFO introduces a novel *Atomized Chain-of-*

*Evaluation* framework. This framework first decomposes omni-condition alignment into a sequential chain of fine-grained, disentangled *atomic evaluation units (AEUs)* across two hierarchical levels, *i.e.*, the local concrete component and global abstract attribute levels. Second, each AEU is categorized into a modality-relevance class, specifying whether its alignment is governed by visual conditions, textual conditions, or their integrated dual-modality combination. Next, UFO further quantifies the alignment score of each AEU either via general Visual Question Answering (VQA) queries or specialized function calls, *e.g.*, leveraging ArcFace (Deng et al., 2019) as a dedicated functional call for high-precision identity verification. Finally, all AEU-level alignment scores are aggregated using adaptive importance weighting to yield an interpretable holistic omni-condition alignment score.

Meanwhile, existing benchmarks for subject-driven customization typically contain limited interactions between textual and visual modalities and few conflicting condition pairs, making them insufficient for comprehensive evaluation of complex multi-modal alignment. To address this gap, we propose **UFO-Bench**, a dedicated benchmark tailored for subject-driven generation under conflicting multi-modal conditions. UFO-Bench comprises 86 reference images across 7 major categories (*e.g.*, humans, rigid objects, anime characters). Crucially, we design a hierarchical prompting strategy for each category, incorporating 3 tiers of editing difficulty with 81.97% conflicting condition pairs. Spanning from simple background shifts to complex edits that simultaneously modify local accessories and global styles, this strategy explicitly tests the ability of evaluation metrics to distinguish between text-guided attribute modifications and core identity preservation.

Our contributions are summarized as follows:

- **Conceptual contribution.** We propose a novel formulation for multi-modal image evaluation, defining consistency assessment as atomic, condition-aware reasoning under free-form multi-modal conditioning.

- **Technical contribution.** We introduce UFO, a chain-of-evaluation framework that integrates fine-grained atomic decomposition, condition-aware VLM evaluation, and expert tools into a unified metric.

- **Benchmark contribution.** We construct UFO-Bench, a comprehensive benchmark featuring diverse subject categories and hierarchical editing scenarios, addressing the lack of fine-grained evaluation testbeds for subject-driven generation.

- **Experimental contribution.** Extensive experiments demonstrate that UFO achieves the highest correlation with human evaluation preferences, improving over existing metrics by 15.25% on average and showing particular advantages in complex scenarios.

## 2. Related Work

### 2.1. Subject-Driven Image Generation

Recent advances in subject-driven image generation focus on creating unified models that enable diverse generation and editing tasks. For example, OmniGen2 (Wu et al., 2025b) separates the decoder and image tokenizer, which supports text-to-image generation, image editing, and context-aware synthesis. In parallel, UNO (Wu et al., 2025c) applies a model-data co-evolution strategy to guide high-resolution data generation and ensure semantically context-consistent outputs. Likewise, the closed-source Nano Banana (DeepMind, 2025) can jointly process text and image inputs within a single framework, thereby supporting both image generation and editing tasks. In the field of open-source models, Qwen-Image (Wu et al., 2025a), built on the backbone of multimodal large language models, has achieved high-quality and flexible subject-aware generation and editing capabilities. BAGEL (Deng et al., 2025a) introduces a Mixture-of-Transformer-Experts (MoT) architecture that models multimodal understanding and generation separately. It enables collaboration through a shared self-attention mechanism, improving cross-modal interaction efficiency and generation consistency.

### 2.2. Subject-Driven Image Generation Benchmarks

In subject-driven image generation research, early efforts mainly focused on benchmarks for general editing tasks. For instance, DreamEditBench (Li et al., 2023) and ImagenHub (Ku et al., 2023) provide standardized datasets and protocols for subject-driven editing, serving as foundational references. Subsequently, evaluations extended to multi-task and 3D scenarios, such as DreamBooth3D (Raj et al., 2023), which adapts subject-driven methods to geometrically consistent generation. Recognizing the limitations of metric-based evaluations, recent works have shifted towards human-aligned assessments. DreamBench++ (Peng et al., 2024) incorporates strong VLMs to approximate human judgment. To address the complexity of customization tasks, DSH-Bench (Wang et al., 2025b) introduces a hierarchical taxonomy with varying difficulty levels (easy/medium/hard), enabling more detailed performance diagnosis. Similarly, focusing on identity preservation, Beyond the Pixels (Singhania et al., 2025) proposes using hierarchical VLM prompting (e.g., feature decision trees) to move beyond superficial similarity scores. However, significant gaps remain. Existing approaches typically evaluate textual and visual conditions in isolation, failing to capture the inherent conflict between identity preservation and complex semantic modifications. They lack a unified framework to reason about the trade-offs in free-form generation, whereas our approach is explicitly designed to assess these conflicting multi-modal interactions through an atomized, condition-aware paradigm.

### 2.3. Evaluation Metrics

**Objective Evaluation Metrics.** Early automated evaluation metrics of text-to-image generation primarily focused on visual quality, such as the Inception Score (IS) (Salimans et al., 2016) and the Frechet Inception Distance (FID) (Heusel et al., 2017). As research progressed, alignment between generated images and textual conditions became more important. Reference-based metrics (Papineni et al., 2002) and reference-free embedding-based approaches such as CLIPScore (Hessel et al., 2021) are introduced to assess text–image consistency. Recently, self-supervised vision representations, notably DINO (Zhang et al., 2022), have been adopted for evaluation. These demonstrate improved sensitivity to object-level and fine-grained semantic differences in image generation and editing tasks.

**LLM-based Evaluation.** In recent years, research has gradually introduced large language models (LLMs) as evaluators.The representative method, VIEScore (Ku et al., 2024), uses a multimodal large language model to decompose semantic consistency and perceptual quality into fine-grained sub-criteria for itemized scoring. However, its evaluation remains limited to the image level. In contrast, Dream-Bench++ (Peng et al., 2024) focuses on personalized image generation tasks and explicitly separates evaluation into two independent dimensions: Image–Image Concept Preservation and Text–Image Prompt Following, which measure the retention of subject features and the quality of instruction execution, respectively. However, this evaluation paradigm relies on absolute scoring of individual-generated results and still has certain limitations in capturing fine-grained quality differences and complex human preferences.

## 3. Method

In this section, we first present a brief analysis of the limitations of existing customization benchmarks, *i.e.*, the insufficient variety of interaction types between textual and visual conditions, making them unable to evaluate the capability of existing customization models in handling multi-modal conditional inputs. To address the gaps, we introduce UFO-Bench in Section 3.1, a dedicated benchmark designed to holistically evaluate the performance of existing customization models under the diverse mutual effects of textual and visual conditions. We then provide a detailed elaboration of **UFO**, the **UniFied Omni**-condition alignment evaluation framework in Section 3.2.

### 3.1. UFO Benchmark

A significant limitation of existing customization benchmarks lies in their **insufficient variety of interaction types** between textual and visual conditions. Most existing benchmarks primarily focus on simple background shifts or sin-

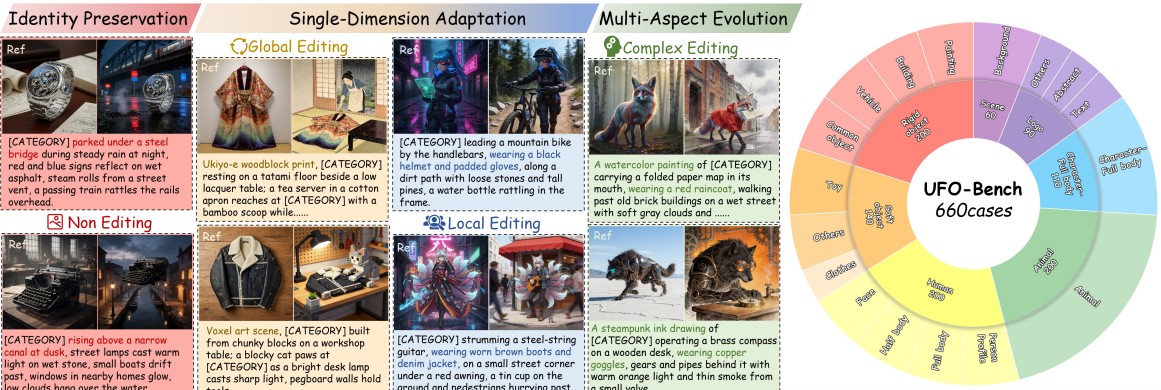

*Figure 2.* **Overview of UFO-Bench.** The benchmark comprises 660 diverse cases spanning hierarchical categories—from rigid objects to full-body characters—and organizes four editing paradigms (Non-Editing, Global Editing, Local Editing, Complex Editing) into three tiers of editing difficulty.

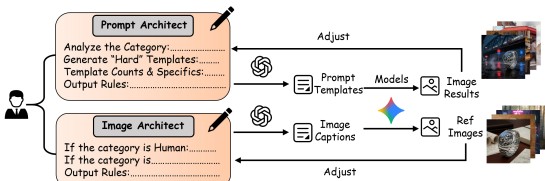

*Figure 3.* **Overview of UFO-Bench construction.** We employ an iterative data preparation pipeline that utilizes LLMs for prompt template generation and caption adjustment to ensure **high-quality benchmark construction**.

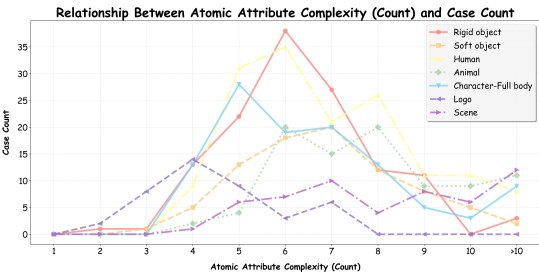

*Figure 4.* Distribution curves of atomic attribute complexity and corresponding case counts for different object categories in UFO-Bench

gular attribute changes, failing to reflect the complexity of **free-form generation** where models must navigate **conflicting multi-modal conditions**. Specifically, they often overlook scenarios requiring the simultaneous execution of holistic style transformations and local interactive modifications. As illustrated in Figure 2, this lack of diversity restricts the comprehensive evaluation of a model's ability to maintain identity preservation while adhering to complex textual instructions. To address these gaps, we introduce **UFO-Bench**, a dedicated benchmark designed to evaluate customization models under diverse and challenging mutual effects of textual and visual conditions.

UFO-Bench is constructed to evaluate the proficiency of generative models in balancing subject-driven image generation with complex textual editing. As illustrated in Figure 2, the benchmark encompasses seven core subject categories—Rigid Object, Soft Object, Human, Full-body Character, Animal, Logo, and Scene—featuring a balanced distribution of simple and complex samples. As shown in Figure 3, we employ an iterative data preparation pipeline utilizing LLMs to ensure high-quality, logically coherent prompts. The design of UFO-Bench revolves around four primary editing paradigms: **Non-Editing**, **Local Editing**, **Global Editing**, and **Complex Editing**, each meticulously tailored to assess distinct model capabilities while ensuring the subject's structural and semantic integrity remains intact. The distribution of atomic attribute complexity across these categories is detailed in Figure 4.

**Non-Editing** tasks focus on recontextualizing the subject without altering its intrinsic attributes, thereby testing the model's ability to seamlessly integrate a core subject into intricate and visually vivid environments. These scenarios prioritize natural lighting, textured backgrounds, and behaviorally consistent actions. For instance, prompts may specify a [CATEGORY] "parked under a steel bridge during steady rain at night," with reflections on wet asphalt and atmospheric steam, or "situated above a narrow canal at dusk" amidst glowing windows and drifting boats. These tasks evaluate the robustness of identity preservation when the subject is subjected to diverse and atmospherically demanding conditions.

**Local Editing** necessitates the addition of specific interactive elements or props, requiring high narrative coherence and natural interaction between the subject and its surroundings. The task design emphasizes action diversity and contextual logic: for human or character subjects, this includes granular details such as "worn brown boots" or actions like "strumming a steel-string guitar"; for other subjects, it in-

volves adding logically consistent props like "a tin cup on the ground" or "a water bottle in a bike frame." These prompts assess the model's precision in performing targeted modifications without disrupting the global identity of the core subject.

**Global Editing** focuses on holistic style or material transformations, challenging models to apply consistent artistic or textural properties while retaining the subject's defining features. The benchmark incorporates recognizable genres and material descriptors, such as Voxel Art (built from chunky blocks), Ukiyo-e woodblock prints (resting on tatami floors), and Steampunk ink drawings. Each prompt provides explicit visual cues, ranging from pixelated aesthetics to traditional Japanese techniques, to evaluate how effectively models can translate a subject into diverse artistic languages without sacrificing identifiability.

**Complex Editing** represents the most rigorous tier, requiring the simultaneous execution of both global stylistic transformations and local element additions. These tasks demand that models reconcile multi-layered instructions within a single coherent frame. Examples include a "watercolor painting of [CATEGORY] carrying a folded paper map in its mouth" or a "steampunk ink drawing of [CATEGORY] operating a brass compass while wearing goggles." By forcing models to navigate style consistency, spatial interaction, and identity preservation in parallel, these tasks provide a comprehensive evaluation of a model's instruction-following capabilities and visual reasoning.

### 3.2. Unified omni-Condition Alignment (UFO)

Although our framework, UFO, is generally applied to multi-modal generation in a free-form manner, our work in this paper focuses on a basic and well-defined personalized image generation setting. Accurate evaluation under this fundamental setting serves as a necessary foundation for more complex multi-image and multi-turn generation scenarios. We propose a novel atomized chain-of-evaluation framework for measuring consistency in multi-modal image generation models. Specifically, the framework comprises four stages. First, omni-condition alignment is decomposed into AEUs. Second, UFO categorizes each AEUs into a modality-relevance class, denoting whether visual, textual, or joint multi-modal factors govern its alignment. Third, each AEU is quantified by VQA queries or specialized function calls. Finally, all AEU-level scores are aggregated with adaptive weighting to obtain an interpretable holistic omni-condition alignment score.

#### 3.2.1. AEUs Decomposition

Multi-condition consistency refers to the visual consistency and the textual consistency of an image generation model. These include maintaining the subject's identity, capturing key visual traits from a reference image, and meeting requirements from a text. The goal of this stage is to clearly evaluate by breaking total consistency into small, measurable, weighted semantic components, making evaluation targets clear.

We design a structured semantic scheme to evaluate free-form multi-modal image generation. Specifically, the reference image ($I_{ref}$) and text ($T_{ref}$) are put into a vision-language model (VLM) to guide the analysis. Then, the model jointly reasons over the image and text, examining subject-specific visual features and text editing constraints. Based on its reasoning, the omni-condition alignment is decomposed into a sequential chain of fine-grained, disentangled atomic evaluation units (AEUs) across two hierarchical levels, i.e., the local concrete component and global abstract attribute levels.

#### 3.2.2. Modality-relevance Classification

Under joint multi-modal constraints, different AEUs exhibit different modality dependencies. Therefore, it is necessary to further determine the modality relevance of each AEU after atomic decomposition.

In this stage, the VLM is used to determine the modality-relevance class of each AEU. Specifically, for each AEU, the model assigns a relevance category $r_i$, which can be image-only, text-only, or text-and-image. As a result, each attribute is evaluated against the correct condition type, enabling fine-grained and semantically accurate assessment of multi-condition consistency.

#### 3.2.3. VQA and Function Calls Scoring

To assess the multi-modal consistency of each decomposed AEU more precisely, each unit is converted into a VQA query. During evaluation, the query, reference text, reference image, and generated image are passed to the VLM. Then, the VLM decides whether the image meets the requirement and outputs Yes or No. The result is then mapped to a score:

$$s_i = \begin{cases} 1, & \text{if the answer is Yes,} \\ 0, & \text{if the answer is No.} \end{cases} \qquad (1)$$

Additionally, we introduce dedicated function calls, which integrate multiple specialized metrics to compensate for the limitations of VLMs. For example, when UFO identifies facial identity consistency as an evaluation attribute, it automatically invokes ArcFace for similarity evaluation. The resulting facial attribute score is normalized to a continuous value in $[0, 1]$, where higher scores indicate stronger identity consistency. The score is computed as:

$$s_i = 1 - d_{\text{ArcFace}}(I_{\text{ref}}, I_{\text{gen}}), \qquad (2)$$

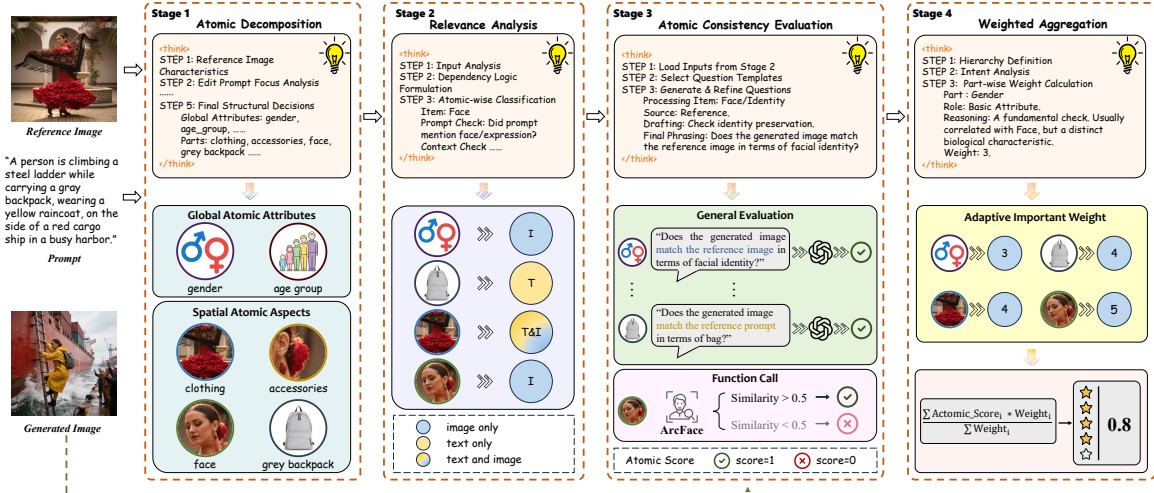

*Figure 5.* **Overall framework of UFO.** UFO performs fine-grained multi-modal evaluation through a four-stage pipeline. Firstly, atomic evaluation units (AEUs) are generated conditioned on the visual and textual inputs. Then each AEU is assigned a modality-relevance class. Subsequently, UFO qualifies the alignment score of each AEU using either VQA or function calls. Finally, adaptive importance weights are applied to aggregate AEU-level scores into a holistic score.

where $d_{\text{ArcFace}}$ denotes the normalized ArcFace feature distance between the reference image $I_{\text{ref}}$ and the generated image $I_{\text{gen}}$.

### 3.2.4. WEIGHTED AGGREGATION

At this stage, all AEU-level alignment scores are aggregated. To enable the UFO to better align with human judgment, each AEU is assigned a non-negative weight $w_i$, where $w_i \in \{1, 2, 3, 4, 5\}$. The weight is determined through VLM-based reasoning over the reference image and textual instruction, based on the importance of subject-specific visual attributes and editing requirements. It reflects the contribution of each AEU to subject preservation and multi-condition consistency. For example, if the reference text mentions a color change, color-related units get more weight. In contrast, when a less important attribute does not meet the requirement, its inconsistency has only a limited impact on the final evaluation score. The final score is calculated for each generated image as the weighted average of its scores:

$$\text{Score}(I_{\text{gen}}) = \frac{\sum_{i=1}^{N} w_i \cdot s_i}{\sum_{i=1}^{N} w_i}, \tag{3}$$

where $s_i$ denotes the consistency score for the $i$-th attribute or component.

This aggregation strategy enables UFO to jointly consider the relative importance of different semantic conditions, resulting in a more interpretable and human-aligned evaluation of multi-modal consistency.

## 4. Experiments

### 4.1. Experimental Setup

**Judge Model.** For automatic evaluation, we adopt GPT-4o as the Judge MLLM, due to its strong multi-modal reasoning capabilities. GPT-4o is used to conduct structured analysis, answer targeted visual questions, and produce AEU-level scores under predefined evaluation criteria. To ensure fair comparison and reproducibility, we use the same GPT-4o model version (2024-05-13) for all evaluations.

**Evaluated Models.** We evaluate a diverse set of both closed-source and open-source multi-modal image generation models. Closed-source models include Nano Banana (Deep-Mind, 2025) and Doubao (specifically utilizing the Seadream4.5 foundation) (ByteDance, 2025). Open-source models include Qwen-ImageEdit-2509, UNO (Wu et al., 2025c), OmniGen2 (Wu et al., 2025b), and Bagel (Deng et al., 2025a). All models are evaluated using their official inference pipelines and recommended hyperparameters.

**Baseline Metrics.** We compare our method against representative automatic evaluation metrics, covering both *LLM-based* and *non-LLM-based* approaches. LLM-based metrics include VIEScore (Ku et al., 2024) and dreambench++ (Peng et al., 2024). Non-LLM-based metrics include CLIP-I (Radford et al., 2021), CLIP-T (Radford et al., 2021), and DINO (Caron et al., 2021).

**Human Evaluation and Alignment.** To assess the consistency between automatic metrics and human judgment, we conduct human evaluation on a randomly sampled subset of 100 test cases, resulting in 600 generated images from six models. For each test case, human annotators are presented

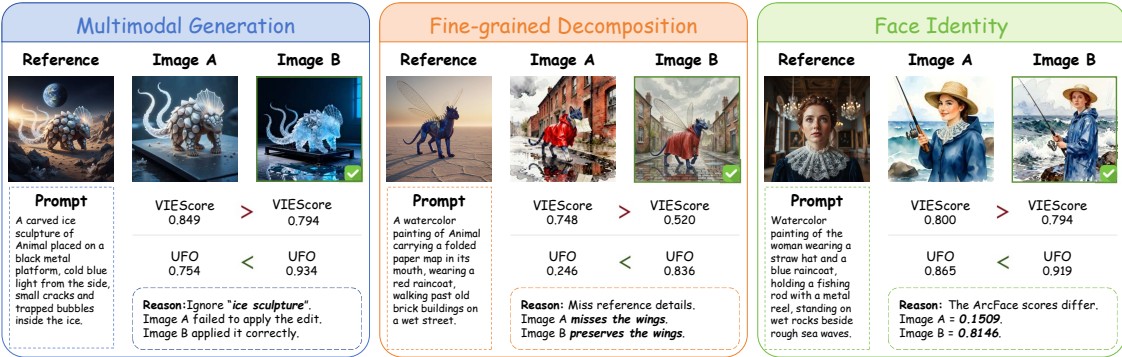

*Figure 6.* **Qualitative comparison with existing methods.** We present three representative cases covering multimodal generation (left, blue), fine-grained decomposition (middle, orange), and face identity evaluation (right, green). UFO produces more reliable evaluation than existing holistic metrics across diverse multi-modal scenarios.

with the six images generated by different models and asked to rank them from 1 to 6 according to overall quality and conditional alignment. Automatic metrics follow the same protocol: for each case, the six model outputs are scored and converted into a relative ranking. We then compute the Spearman rank correlation coefficient between the human ranking and the metric-induced ranking on a per-case basis, and report the average Spearman correlation across all cases. Additionally, we apply Fisher Z-transformation to estimate the average Spearman correlation $\in [-1, 1]$ across models. A higher Spearman correlation indicates stronger agreement with human judgment in relative model comparison.

### 4.2. Main Results

**Quantitative Evaluation on UFO-Bench.** To comprehensively assess the capability of current state-of-the-art models, we conducted an extensive evaluation using the UFO-Bench dataset. The evaluation protocol involved generating outputs for every reference-prompt pair, with each pair sampled four times to mitigate generation randomness, resulting in a total of 2,640 instances per model.

As presented in Table 1, the closed-source model Doubao (2025-12) demonstrates superior performance, achieving the highest total score of 0.7439. It exhibits remarkable stability across both local and global editing tasks, suggesting a robust understanding of fine-grained attribute manipulation. Notably, the open-source model Qwen-Image (2025-12) delivers a competitive performance (total score: 0.7252), surpassing the closed-source Nano-Banana (2025-08) and significantly outperforming other community models. Interestingly, while Qwen-Image excels in non-editing (0.7635) and global editing (0.7836), even slightly surpassing Doubao in these categories, it shows a noticeable performance drop in local editing (0.6532). This discrepancy highlights a common limitation in current open-source diffusion architectures: while they capture global semantic shifts effectively, precise text-guided editing remains a challenge

compared to proprietary commercial models.

**Correlation with Human Judgments.** Following the evaluation protocol described in the Setup, we benchmark UFO against a set of existing metrics. The Spearman rank correlation coefficients are reported in Table 2.

Traditional embedding metrics demonstrate a severe misalignment with human preference in the subject-driven image generation task. Notably, CLIP-T exhibits a negative correlation (-0.1996). This result indicates that focusing solely on global text-image alignment is not a reliable metric for personalized tasks. Similarly, CLIP-I and DINO achieve only weak positive correlations ($\rho \approx 0.3 \sim 0.4$), indicating their limitations in evaluating fine-grained conditional image editing.

Recent MLLM-based metrics (*i.e.*, VIE-Score, Dream-Bench++) show improved alignment ($\rho > 0.55$) by leveraging the reasoning capabilities of GPT-4o. However, they struggle with localized precision. As evidenced in the local editing column, these methods achieve correlations around 0.58 because they lack an explicit mechanism to decouple the editing region from the preservation background.

Figure 6 further presents representative qualitative cases from three perspectives. The left case focuses on conflicting multi-modal conditions, showing that existing metrics may favor visual similarity over correct textual instructions. The middle case shows that fine-grained atomic decomposition better evaluates subject detail preservation, while holistic metrics often overlook subtle local inconsistencies. The right case presents identity-sensitive face evaluation, where expert function calls enable UFO to correctly detect facial identity inconsistencies overlooked by existing metrics.

### 4.3. Ablation Study

To investigate the individual contributions of the proposed components within the UFO framework, we conduct a series

*Table 1.* Quantitative results of latest customization and unified models on **UFO-Bench**, including both closed-source commercial and open-source community variants, evaluated via the omni-condition alignment metric **UFO**. Higher score denotes better performance. We highlight the [best] and [second-best] values for each task.

| Model | Non-Editing | Single-Dim-Editing | | Multi-Dim-Editing | Total Score |
|---|---|---|---|---|---|
| | | Local | Global | | |
| *Closed-Source Commercial Models* | | | | | |
| Doubao (*2025-12*)(ByteDance, 2025) | 0.7360 | 0.7141 | 0.7807 | 0.7475 | 0.7439 |
| Nano-Banana (*2025-08*)(DeepMind, 2025) | 0.6548 | 0.7105 | 0.6739 | 0.6715 | 0.6816 |
| *Open-Source Community Models* | | | | | |
| Qwen-Image (*2025-12*)(Wu et al., 2025a) | 0.7635 | 0.6532 | 0.7836 | 0.7257 | 0.7252 |
| OmniGen2 (*2025-06*)(Wu et al., 2025b) | 0.5126 | 0.4214 | 0.3885 | 0.3980 | 0.4235 |
| BAGEL (*2025-05*)(Deng et al., 2025a) | 0.2363 | 0.3581 | 0.4484 | 0.4719 | 0.3866 |
| UNO (*2025-04*)(Wu et al., 2025c) | 0.4029 | 0.4381 | 0.4924 | 0.4746 | 0.4549 |

*Table 2.* Quantitative comparison of alignment with human judgments against other automatic metrics, demonstrating that our proposed UFO achieves the highest correlation with human evaluations. We highlight the [best] and [second-best] values for each metric.

| Metric | Non-Editing | Single-Dim-Editing | | Multi-Dim-Editing | Total |
|---|---|---|---|---|---|
| | | Local | Global | | |
| CLIP-I(Radford et al., 2021) | 0.2141 | 0.3028 | 0.4795 | 0.4358 | 0.3910 |
| CLIP-T(Radford et al., 2021) | -0.2299 | -0.1580 | -0.3946 | -0.2141 | -0.1996 |
| DINO(Caron et al., 2021) | 0.1270 | 0.1281 | 0.4791 | 0.3952 | 0.3084 |
| VIE-Score SC(Ku et al., 2024) | 0.6948 | 0.5819 | 0.4591 | 0.6788 | 0.5752 |
| VIE-Score PQ(Ku et al., 2024) | -0.1158 | -0.0349 | 0.1538 | 0.0565 | 0.0192 |
| VIE-Score O(Ku et al., 2024) | 0.6715 | 0.5889 | 0.4538 | 0.6735 | 0.5663 |
| DreamBench++(Peng et al., 2024) | 0.7262 | 0.5724 | 0.5884 | 0.5744 | 0.5977 |
| **UFO (Ours)** | **0.8034** ($\pm$ 10.63%) | **0.6837** ($\pm$ 16.10%) | **0.6295** ($\pm$ 6.99%) | **0.7241** ($\pm$ 6.67%) | **0.6889** ($\pm$ 15.26%) |

*Table 3.* Ablation study on the correlation of different components in the proposed UFO with human preferences.

| Setting | Correlation with Human Preference |
|---|---|
| UFO *w/o AEUs Decomposition* | 0.5752 |
| UFO *w/o weighted aggregation* | 0.6253 |
| **UFO (full setting)** | **0.6889** |

of ablation studies. The results, summarized in Table 3, demonstrate that both the atomic decomposition strategy and the weighted aggregation mechanism are critical for achieving high alignment with human preferences.

**Effect of Weighted Aggregation.** To validate the necessity of preference-aware weighting, we replace the adaptive weighting scheme with a uniform approach, *i.e.*, setting all reward weights to 1. As shown in the second row of Table 3, this leads to a decrease in the correlation coefficient from 0.6889 to 0.6253. This performance drop suggests that without specific preference weighting, the model treats

all attributes equally. Consequently, violations in trivial attributes may overly penalize the overall score, while successful alignment with important visual characteristics or critical textual instructions is not sufficiently rewarded.

**Effect of Atomic Decomposition.** In the setting "w/o AEUs Decomposition", the generated image is fed directly into the VLM to produce a single holistic score. This ablation results in a significant performance degradation (0.6889 $\rightarrow$ 0.5752), indicating that holistic scoring struggles to capture complex visual details, whereas our atomic decomposition strategy enables more precise and robust evaluation.

## 5. Conclusion

We introduce UFO, a unified framework for omni-condition alignment that evaluates atomic modality-relevance units and aggregates them into a holistic score. We further propose UFO-Bench, a benchmark spanning diverse subjects and hierarchical editing scenarios. Extensive experiments show that UFO achieves stronger alignment with human judgment than existing metrics, particularly under complex multi-modal conditions.

## Acknowledgements

This research is supported by the National Natural Science Foundation of China (Grant No. 623B2094) and the Fundamental and Interdisciplinary Disciplines Breakthrough Plan of the Ministry of Education of China (Grant No. JYB2025XDXM103).

## Impact Statement

This paper presents work whose goal is to advance the field of Machine Learning. There are many potential societal consequences of our work, none which we feel must be specifically highlighted here.

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
