# A. Additional Benchmark Visualizations

Figures 6 and 7 provide supplementary visualizations of UFO-Bench.

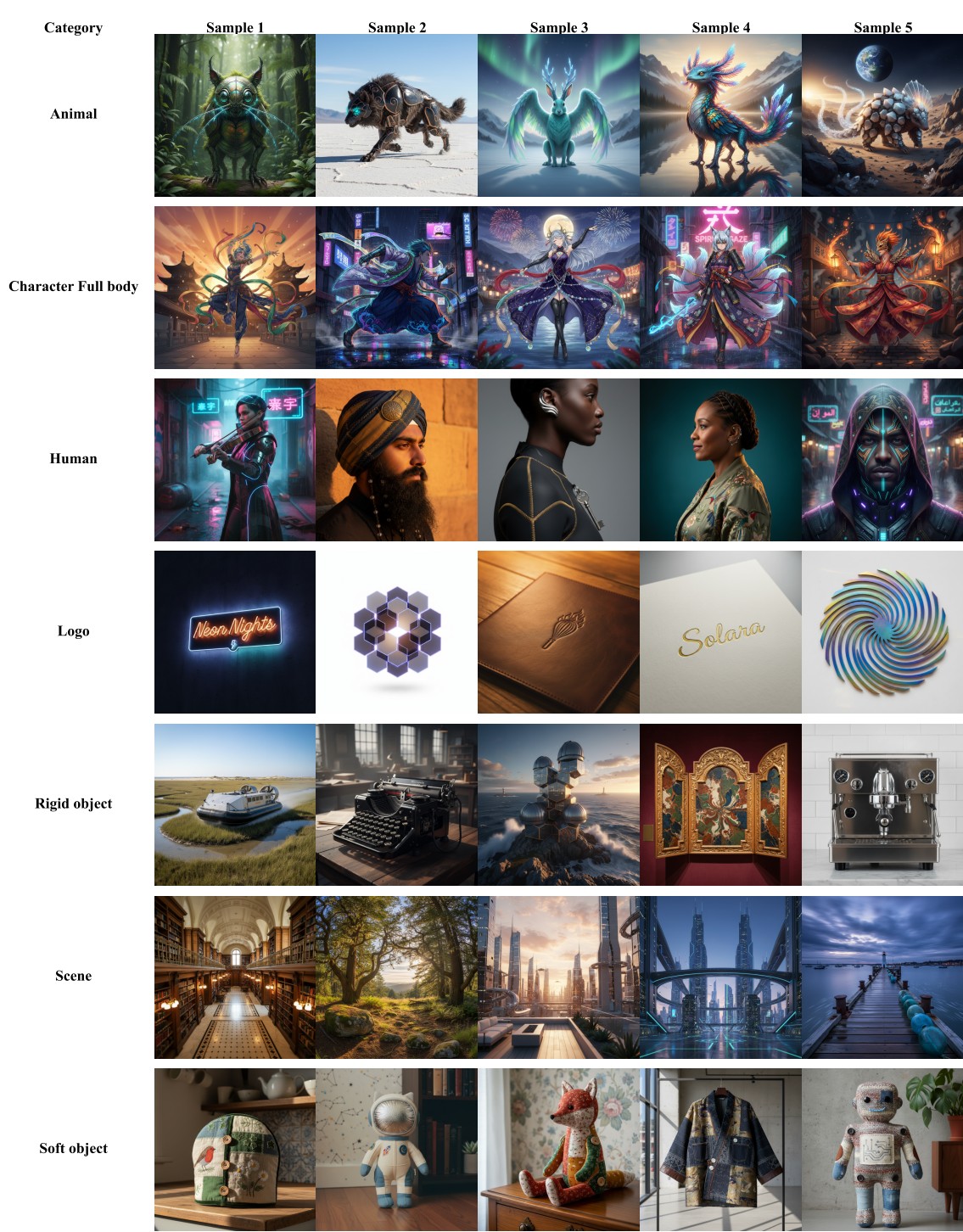

*Figure 6.* Reference image samples from UFO-Bench illustrating the diversity of visual categories and structural complexity covered by the benchmark.

| Category | Subtypes | Non-Editing Templates | Global-Editing Templates | Local-Editing Templates | Complex-Editing Templates |
|---|---|---|---|---|---|
| Logo | Text, Abstract, Others | 1. the [CATEGORY] displayed on a giant LED billboard above a busy crosswalk at night, rain streaking down the screen, red and blue signs reflecting on wet pavement, umbrellas moving through the crowd, steam drifting from a street vent, and a late bus pulling up under bright white lamps 2. the [CATEGORY] printed on a taut white vinyl banner stretched across scaffolding at a building renovation site in the sunlight, dust hanging in the air, workers in yellow vests carrying plywood, a yellow crane swinging a pallet, and loose cables tied to metal poles | 1. ukiyo-e woodblock print of the [CATEGORY] on textured rice paper, soft black ink lines with flat red and blue fills, slight misregistration at the edges, a faint wood grain pattern, and a simple off-white border 2. voxel art rendering of the [CATEGORY] built from small cube blocks, clean edges with bright primary colors, subtle shadows on a plain gray tabletop, and a soft studio light from the left 3. gold foil embossed version of the [CATEGORY] pressed into black leather, sharp raised edges, fine grain texture catching warm light, with tiny scuffs and worn corners around the imprint | 1. stage technician pushing a fog machine across a dark concert stage while the [CATEGORY] is projected onto the haze behind, blue LED bars glowing along the floor, loose gaffer tape on black cables, and small dust specks in the light 2. sign maker in an orange vest applying a large adhesive of the [CATEGORY] onto a glass storefront with a rubber squeegee, soap water running down the pane, bright noon sun casting clear shadows, and passersby reflected in the glass 3. barista stamping the [CATEGORY] onto white paper cups with a red ink stamp at a wooden counter, stacks of cups on the side, a stainless steel machine humming in the back, and warm yellow bulbs above the bar | 1. neon tube sign of the [CATEGORY] in hot pink mounted on a black metal grid, power supply boxes bolted behind it, while an electrician on a gray ladder tightens a bracket with a wrench, loose wires coiled on the floor and a faint red glow on the wall 2. ice relief carving of the [CATEGORY] set on a low black stage riser under cold blue light, sharp frosted edges with clear faces, while a stagehand in gloves sprays water from a bottle to smooth the surface, thin puddles forming at the base |
| Animal | - | 1. [CATEGORY] sprinting along wet cobblestones under a light rain, street lamps casting warm yellow light, puddles mirroring red and blue shop signs, a tram sliding by in the background with tiny sparks at the wire overhead. 2. [CATEGORY] climbing a mossy rock beside a fast river in a narrow canyon at sunset, orange light catching the spray, dark pine trees lining the cliffs, cool mist drifting over the water and small ripples breaking against the stones. | 1. An ukiyo-e woodblock print of [CATEGORY] crossing a wooden bridge over a calm river, cherry blossoms falling, flat colors with bold outlines, paper grain visible and gentle ripples rendered with simple curved lines. 2. A voxel art diorama of [CATEGORY] moving through a blocky snow field with cube trees, bright plastic texture, simple shadows on a white ground, tiny stacked cubes forming tracks behind it. 3. A carved ice sculpture of [CATEGORY] placed on a black metal platform, cold blue light from the side, small cracks and trapped bubbles inside the ice, melting water pooling on the floor with faint reflections. | 1. [CATEGORY] trotting along a gravel trail, tugging a small red wagon by a rope handle, wearing a bright blue bandana, in a pine forest at dawn with thin fog between the trunks and damp needles underfoot. 2. [CATEGORY] pawing at a soccer ball and pushing it toward a rusty goal, wearing a striped green jersey, in a muddy field after fresh rain with bright stadium lights and wet grass shining. 3. [CATEGORY] resting by a campfire while guarding a canvas backpack, wearing a reflective harness, at a lakeside campsite with a pitched tent, sparks rising into the night and faint stars between light clouds. | 1. A watercolor painting of [CATEGORY] carrying a folded paper map in its mouth, wearing a red raincoat, walking past old brick buildings on a wet street with soft gray clouds and shallow puddles along the curb. 2. A steampunk ink drawing of [CATEGORY] operating a brass compass on a wooden desk, wearing copper goggles, gears and pipes behind it with warm orange light and thin smoke from a small valve. |
| Human | A Person's Profile, Face, Half body, Full body | 1. [CATEGORY] sprinting through heavy rain on a tight city street at night, red and blue lights glow on wet pavement, buses spray water as they pass, steam from a street vent blows across the road, wind pushes loose flyers off a brick wall, face lit by mixed colors from the signs and a passing bus. 2. [CATEGORY] seated on a weathered wooden pier at sunrise, waves tap the posts, small fishing boats drift near the harbor mouth, coils of rope and old nets lie beside worn planks, the sky shifts from pink to light orange, sunlight warms the water and the subject watches the horizon. | 1. Ukiyo-e woodblock print showing [CATEGORY] walking beside a river lined with pink cherry trees, tall paper lanterns hang from wooden poles, flat colors and bold black lines, ripples in the water reflect lantern light, simple clouds stretch across the evening sky. 2. Voxel art depiction of [CATEGORY] crossing a small square in front of a brick building, blocky benches and a blocky bike rack stand nearby, clear blue sky above, strong square shadows fall across gray tiles and a short set of steps. 3. White marble statue of [CATEGORY] placed inside a museum hall, smooth stone surface with small chisel marks, bright light from a skylight above, long dark shadows across the floor, metal railings and an information stand set a few feet away. | 1. [CATEGORY] climbing a steel ladder while carrying a gray backpack, wearing a yellow raincoat, on the side of a cargo ship in a busy harbor, waves slap the hull and workers shout from the dock. 2. [CATEGORY] serving hot coffee from a steel pot into paper cups, wearing a green apron and a white shirt, behind a wooden counter in a crowded cafe, steam fogs a small window near the espresso machine. 3. [CATEGORY] steering a small inflatable boat with one hand and holding a handheld radio with the other, wearing an orange life vest and black boots, on a dark lake at night, a lantern tied to the bow throws warm light on the waves. | 1. Watercolor painting of [CATEGORY] wearing a straw hat and a blue raincoat, holding a fishing rod with a metal reel, standing on wet rocks beside rough sea waves, soft washes of blue and gray with visible brush marks and paper texture. 2. Ice sculpture of [CATEGORY] wearing a thick scarf and gloves, holding a metal lantern raised at chest level, set in a winter town square at night, street lamps give white light, frost on stone tiles and fine cracks run through the clear ice. |
| Soft object | Clothes, Toy, Others | 1. Rainy night street market with wet pavement reflecting red and blue signs; a kid with a plastic bag reaches at [CATEGORY] on a metal rack while steam drifts from a food cart, streetlights flicker, and a stray cat waits under a bench. 2. Early morning bedroom with soft sun through thin curtains and dust in the air; a small dog sits at [CATEGORY] placed at the edge of a neatly made bed, a ceiling fan hums, posters on the wall, and wooden floorboards show light scuffs. | 1. Ukiyo-e woodblock print, [CATEGORY] resting on a tatami floor beside a low lacquer table; a tea server in a cotton apron reaches at [CATEGORY] with a bamboo scoop while sliding doors open to a small garden in the rain. 2. Voxel art scene, [CATEGORY] built from chunky blocks on a workshop table; a blocky cat paws at [CATEGORY] as a bright desk lamp casts sharp light, pegboard walls hold tools, and a small plant in a square pot sits by the edge. 3. Crystal ice sculpture of [CATEGORY] set on a black stone pedestal, tiny chips on the base; a bartender in a white shirt wipes water at [CATEGORY] with a clean towel, back bar shelves glow warm amber while cold mist bugs the counter. | 1. Inside a mountain cabin, a hiker wearing a yellow raincoat and a heavy backpack ties a paper tag at [CATEGORY] set on a rough wooden table; a folded map, a metal compass, and a lit lantern add warm light as rain taps on the window. 2. Under a living room blanket fort, a child in striped pajamas holding a small flashlight whispers at [CATEGORY] while string lights glow, pillows stack as walls, and crayons roll across a low coffee table. 3. Tailor workshop with chalk dust in the air; a tailor wearing round glasses uses a silver needle and blue thread at [CATEGORY] spread on a pine workbench, with pattern paper, pins in a tomato pincushion, and a bright desk lamp. | 1. Watercolor painting with loose brush strokes, [CATEGORY] placed on a city sidewalk; a cyclist wearing a red scarf kneels at [CATEGORY] while fixing a slipped chain, rain leaves small puddles, shop windows glow, and soft gray clouds hang low. 2. Steampunk digital art, [CATEGORY] reimagined with leather panels and brass rivets; a mechanic with goggles tightens a valve at [CATEGORY] using a small wrench as pipes vent steam, gears turn on a wall, and warm orange light fills the room. |
| Character-Full body | - | 1. [CATEGORY] sprinting along a wet city street at night, rain bouncing off the pavement, bright store signs reflecting in shallow puddles, headlights throwing long shadows, steam drifting from a subway grate, wind pushing loose flyers across the sidewalk. 2. [CATEGORY] climbing a rocky hillside at dawn, hands braced on rough stone, thin fog rolling over short grass, the sun rising behind low clouds, a radio tower on the ridge, a flock of birds breaking from a tree as wind whistles across the slope. | 1. Ukiyo-e woodblock print of [CATEGORY] walking across a wooden bridge over a blue river, cherry trees with pink petals along the banks, flat colors with bold black outlines, a small village in the sunlight with paper lanterns. 2. Voxel art of [CATEGORY] hiking through a blocky pine forest, cube rocks and sharp edges, bright green grass cubes, a square sun over stepped hills, hard pixel shadows and simplified forms. 3. Polished marble statue of [CATEGORY] posed on a museum plinth, cool white stone with fine chisel marks, soft gallery lights from above, faint reflections on a dark floor, a rope barrier and wall plaques behind. | 1. [CATEGORY] running while carrying a weathered backpack, wearing a yellow raincoat, through a crowded train station with wet tiles, departure boards flickering, and overhead speakers crackling. 2. [CATEGORY] strumming a steel-string guitar, wearing worn brown boots and a denim jacket, on a small street corner under a tin cup on the ground and pedestrians hurrying past. 3. [CATEGORY] leading a mountain bike by the handlebars, wearing a black helmet and padded gloves, along a dirt path with loose stones and tall pines, a water bottle rattling in the frame. | 1. Watercolor painting of [CATEGORY] wearing a red climbing harness, gripping a chalked carabiner, on a sunlit cliff with green trees below, loose brush strokes and light washes, paper texture visible. 2. Ice sculpture of [CATEGORY] holding a metal lantern with a warm orange glow, wearing a thick scarf carved with simple ridges, on a snowy town square at night, blue lights shining through clear ice and small puddles forming around the base. |
| Scene | Background | 1. [CATEGORY] at dawn, fog rolling over wet cobblestone streets, long shadows from old brick buildings, yellow window lights reflecting in puddles, paper flyers stuck to a wooden fence, birds lifting from roof edges into a pale sky. 2. [CATEGORY] under heavy rain at night, neon signs in red and blue blinking over metal shop grates, water streaming along gutters, tangled wires crossing the street, steam drifting from a vent, faint glow on cracked concrete walls. | 1. [CATEGORY] rendered as a ukiyo-e woodblock print with flat colors and bold outlines, fine rain lines across stone paths, lanterns casting warm yellow on dark water, pine trees simplified into sharp shapes, waves drawn as curved bands along a harbor. 2. [CATEGORY] built as a voxel diorama with blocky houses, blocky trees, and blocky cars, crisp edges on gray roads, bright green lawns, a blue sky with square clouds, simple shadows under eaves and benches. 3. [CATEGORY] formed from clear ice blocks, rough chisel marks on frozen steps, cold blue light passing through thick walls, frost gathering on metal rails, thin cracks catching white light along the edges. | 1. [CATEGORY] with a street vendor grilling corn on a metal cart under a red umbrella, smoke rising and mixing with drizzle, two kids holding paper cups waiting in line, greasy paper bags stacked on a wooden crate. 2. [CATEGORY] with a team of workers in orange vests hauling a steel beam from a flatbed truck, one worker guiding with gloved hands, sparks from a welding torch lighting the underside of a scaffold, traffic cones lined along cracked asphalt. 3. [CATEGORY] with a cyclist wearing a green rain jacket weaving around puddles, a yellow taxi braking at a crosswalk, a dog tugging a leash near a mural, wet pavement mirroring blue shop lights. | 1. [CATEGORY] rendered as a ukiyo-e woodblock print, a fisherman wearing a straw hat pulling a net from a small wooden boat near a pier, simple wave patterns circling the hull, flat clouds layered above, red seals in the corner like artist marks. 2. [CATEGORY] built as a voxel scene, a rescue helicopter hovering over a rooftop while a person in an orange jacket waves a bright flag, blocky dust blowing from the edge, square windows lit yellow, a ladder leaning against a brick wall. |
| Rigid object | Painting, Building, Vehicle, Common object | 1. [CATEGORY] rising above a narrow canal at dusk, street lamps cast warm light on wet stone, small boats drift past, windows in nearby homes glow, low clouds hang over the water. 2. [CATEGORY] parked under a steel bridge during steady rain at night, red and blue signs reflect on wet asphalt, steam rolls from a street vent, a passing train rattles the rails overhead. | 1. Ukiyo-e woodblock print of [CATEGORY] in rain with strong black lines and flat colors, paper texture visible, lantern light along a river bank. 2. Voxel art version of [CATEGORY] built from small cubes with bright colors and sharp edges, placed on a plain gray base with soft shadow. 3. Marble statue of [CATEGORY] carved from white stone with light chisel marks, set on a square pedestal in a gallery with cool overhead lights. | 1. A mechanic wearing a gray jumpsuit kneels at [CATEGORY], tightening a loose panel with a ratchet, a small toolbox open on wet asphalt under bright streetlight. 2. A museum curator with white gloves stands at [CATEGORY], dusting its surface with a soft brush, quiet hall around them, polished wood floor mirrors the scene. 3. A child with a red backpack sits at [CATEGORY], sketching it in a small notebook, pigeons peck at crumbs nearby, afternoon sun breaks through thin clouds. | 1. A watercolor illustration of [CATEGORY] with light washes and soft lines, a street vendor in a blue apron pours hot tea at [CATEGORY] while rain makes small rings in puddles on the ground. 2. A marble statue of [CATEGORY] with fine chisel lines and a slight matte finish, a small ginger cat rubs its side at [CATEGORY] in a quiet courtyard with green vines on brick walls. |

*Figure 7.* Prompt Templates of UFO-Bench.