# OpenReview forum: "UFO: Chain-of-Evaluation for Omni-Condition Alignment in Multi-Modal Image Generation"
_ICML.cc/2026/Conference — ICML 2026 regular_

### Official Review · Reviewer_Fxnm · 2026-03-09

**Soundness:** 4
**Presentation:** 3
**Significance:** 4
**Originality:** 3
**Overall Recommendation:** 5
**Confidence:** 4

**Summary:**

This paper introduces UFO, a unified evaluation framework for subject-driven multi-modal image generation that addresses the core limitation of existing metrics: evaluating textual and visual condition alignment in isolation.

Alongside UFO, the authors contribute UFO-Bench — a benchmark of 660 cases spanning 7 subject categories and 4 editing paradigms of increasing complexity.

Experiments demonstrate that UFO achieves higher correlation with human judgments than existing baselines, with particularly strong gains on local editing tasks.

**Compliance With Llm Reviewing Policy:**

Affirmed.

**Key Questions For Authors:**

See weaknesses.

**Limitations:**

No. The paper does not include a dedicated limitations section. The authors should explicitly discuss: (1) the dependence on a proprietary, paid API (GPT-4o) and its implications for reproducibility and cost; (2) the potential for the AEU decomposition step to introduce errors that cascade through the pipeline; (3) the fact that the framework has only been validated on subject-driven customization and may not generalize to other multi-modal generation tasks without re-evaluation.

**Strengths And Weaknesses:**

**Strengths:**

The study provides a well-motivated theoretical and empirical foundation, demonstrating that isolated evaluation leads to systematic errors.

By establishing the UFO-Bench with its principled, hierarchical editing difficulty tiers, the work addresses a critical bottleneck in generative research, offering a framework that more faithfully aligns with human judgment and provides significant downstream value for multi-condition alignment benchmark.

**Weaknesses:**

1. Concerns still exist with respect to the reproducibility, the benchmark provided herein, or the open-source nature of the evaluation tools in this paper. If the code or evaluation tools cannot be made open-source, it would significantly undermine the contribution of this work.

2. The paper's contributions section refers to the framework as "OminiMM" and the benchmark as "OminiBench," while the rest of the paper consistently uses "UFO" and "UFO-Bench." This inconsistency suggests the paper may have been hastily revised and undermines clarity. Reviewers and readers may be confused about whether these are distinct systems or the same thing under different names.

3. The adaptive importance weighting is a key differentiator, but Section 3.2.4 does not adequately explain how weights are assigned. Figure 5 shows a "STEP 3: Part-wise Weight Calculation" inside a thinking block, suggesting GPT-4o generates the weights, but the prompting strategy, the scale of weights, and what prevents arbitrary or inconsistent assignments are not described.

4. AEU-level scoring is binary (0 or 1 via VQA). This granularity may be too coarse for subtle alignment differences. A partial-credit scheme or confidence-calibrated score could improve sensitivity.

5. The entire framework depends on a single proprietary model (GPT-4o, version 2025-05-13) for AEU decomposition, modality classification, VQA scoring, and weight assignment. This raises concerns about cost, API access, long-term reproducibility, and sensitivity to future model updates. No analysis is provided on how UFO performs with alternative judge models.

6. Only 100 test cases are used for human evaluation, with 3 annotators implied but not described in detail. Inter-annotator agreement is not reported, making it difficult to assess the reliability of the human preference signal used as ground truth.

7. The paper does not discuss cases where UFO disagrees with human judgment or where AEU decomposition fails (e.g., hallucinated AEUs, incorrect modality classification). Such analysis would strengthen confidence in the method.

---

> ### Author Rebuttal · Authors · 2026-03-31
>
> We thank the reviewer for the thoughtful review. We respond to each concern in detail:
>
> **R1:** We'd like to clarify that the code&benchmark will be released for reproducibility.  Furthermore, the function calls in our evaluation pipeline (e.g., ArcFace for face analysis) rely on publicly available open-source tools. Due to the rebuttal time limit, we provide the comparison with Qwen3-4B in the following table and we will include more open- and closed-source MLLMs (like Gemini) in the revision. We compare the human alignment between our framework and VIE-Score.
>
> |                                   | ours   | VIE-Score |
> | --------------------------------- | ------ | --------- |
> | GPT4o                             | 0.6889 | 0.5663    |
> | Qwen3-4B                          | 0.2762 | 0.1326    |
> | Degree of human alignment decline | 59.90% | 76.58%    |
>
> As shown in the table, the VLM in our metric replaced from GPT-4o to Qwen3-4B leads to a **59.90%** drop in human alignment. In contrast, when VIE-Score uses Qwen3-4B to compute,  the consistency relative to GPT-4o decreases by **76.58%**, indicating that our metric is **more** **robust** **to evaluator changes**.
>
> **R2:** We clarify that this is a typo and **does not affect any experimental results, data, or conclusions**. We have **carefully and thoroughly verified all experiment-related data and analyses (we pay our most attention to these parts) to ensure their full correctness**. We appreciate your careful review and will correct this typo in the revised manuscript.
>
> **R3:** The importance weights are integers ranging from 1 to 5, assigned by the VLM according to visual attributes and editing requirements, where elements critical to editing or identity preservation receive higher weights and secondary or less important structures receive lower weights; these weights remain stable across runs, and we will clarify the prompting strategy and assignment criteria in the revision.
>
> **R4:** We thank for the suggestion. To investigate this concern, we conduct additional experiments with three scoring schemes: (1) the original binary (0/1) AEU scoring, (2) a partial-credit scheme (1–5), and (3) a confidence-calibrated score derived from the VLM’s confidence for “yes/no” predictions. We evaluate these variants by measuring their alignment with human judgments. The experiments are conducted using Qwen-based evaluators, and the results are reported in table below:
>
> |          | Human alignment | latecy(s) |
> | -------- | --------------- | --------- |
> | Score5   | 0.3988          | 4.5362    |
> | P_yes/no | 0.2923          | 50.0891   |
> | Score0/1 | 0.2874          | 50.0891   |
>
> As shown in the table, using a 1–5 scoring scale yields better alignment with human. We will further refine our scoring scheme in revison. Importantly, we'd like to clarify that **it does not affect our core contribution**: We sincerely appreciate the reviewer’s thoughtful suggestions and will **include them in the revised manuscript**.
>
> **R5:**   We'd like to clarify that the UFO is a framework, which is **applicable to a wide range of VLMs** and does not rely on any proprietary model. Experiments with other MLLMs (e.g., Qwen3-4B) show consistent results, demonstrating that our framework remains effective when using **open-source VLMs**. Here are the results:
>
> |                                   | ours   | VIE-Score |
> | --------------------------------- | ------ | --------- |
> | GPT4o                             | 0.6889 | 0.5663    |
> | Qwen3-4B                          | 0.2762 | 0.1326    |
> | Degree of human alignment decline | 59.90% | 76.58%    |
>
> As shown in the table, the VLM in our metric replaced from GPT-4o to Qwen3-4B leads to a **59.90%** drop in human alignment. In contrast, when VIE-Score uses Qwen3-4B to compute,  the consistency relative to GPT-4o decreases by **76.58%**, indicating that our metric is **more robust to evaluator changes**. For reproducibility and long-term accessibility, we will release the code.
>
> **R6:**  We clarify that the size of the human evaluation set follows common practice in recent method papers,  where human evaluations typically use 100–300 randomly sampled examples, while much larger scales are mainly used in benchmark construction. For example, REAL[1] builds its evaluation set by sampling small subsets from existing datasets (e.g., 200 from Birds).For reproducibility, we will release the human evaluation set together with the benchmark.
> [1] REAL: Realism Evaluation of Text-to-Image Generation Models for Effective Data Augmentation
>
> **R7:**  We clarify that the main failure case of UFO stems from the **lack of specialized function calls** for fine-grained evaluation on certain specific dimensions.For example, in style-editing tasks, the absence of a dedicated style classification module may lead to incorrect modality splits or failures in identifying the correct subject entity.We will include representative failure cases in the revised manuscript.

---

> > ### Author Rebuttal · Reviewer_Fxnm · 2026-04-02
> >
> > The author addressed my concerns.

---

> > > ### Author Response · Authors · 2026-04-08
> > >
> > > We greatly appreciate the reviewer’s insightful and positive feedback, which has significantly improved the quality of our work.

---

### Official Review · Reviewer_bvxv · 2026-03-10

**Soundness:** 2
**Presentation:** 2
**Significance:** 3
**Originality:** 3
**Overall Recommendation:** 4
**Confidence:** 4

**Summary:**

This paper proposes an evaluation framework for subject-driven multi-modal image generation, termed UFO, which aims to jointly assess alignment with both text and image conditions. The framework decomposes “omni-condition alignment” into Atomic Evaluation Units (AEUs), assigns each unit a modality-relevance category, and scores them via GPT-4o combined with expert tools, followed by adaptive weighted aggregation. The authors also introduce UFO-Bench, a small benchmark containing 86 reference images and 660 cases, on which the proposed framework achieves higher correlation with human judgments than existing methods.

**Compliance With Llm Reviewing Policy:**

Affirmed.

**Final Justification:**

The authors' rebuttal addresses most of my concerns. Therefore, I decide to raise my rating to weak accept.

**Key Questions For Authors:**

1. The UFO-Bench benchmark needs to be elaborated on. How are the data and cases in UFO-Bench distributed across categories and editing types? Why is the benchmark designed in this particular way? What is the necessity of constructing this benchmark: does it supplement existing datasets/benchmarks , or is it intended to replace them in certain scenarios?
2. The authors need to disclose more detailed experimental and implementation settings about the UFO evaluation framework, and provide stronger evidence of its generalization and effectiveness beyond the self-constructed UFO-Bench.

See the weaknesses as well.

**Limitations:**

No. Although the paper briefly acknowledges the difficulty of topic-driven evaluation, it does not fully discuss the limitations of the proposed framework. Specifically, (i) all results rely on a single closed-source evaluation model (GPT-4o), without exploring the performance of this method under weaker or different MLLMs; (ii) the claim of human alignment is only validated on the authors' own benchmark (UFO-Bench), without discussing the scalability of the framework; (iv) the construction, and potential biases of UFO-Bench are not disclosed in detail. These omissions should be clearly stated.

**Strengths And Weaknesses:**

**Strengths**:
1. The paper proposes a unified evaluation framework (UFO) for subject-driven multi-modal image generation, aiming to jointly assess alignment with both text and image conditions rather than treating them in isolation.
2. The authors construct a small benchmark (UFO-Bench) with multiple editing regimes (non-editing, local, global, complex).
3. Conceptually, the work introduces a modality-aware decomposition of evaluation into atomic units, explicitly categorizing attributes as image-only, text-only, or text-and-image.

**Weaknesses**：
1. The paper introduced a new dataset, yet there is hardly any detailed introduction of UFO-Bench in the main text, and the paper does not evaluate the utility of this benchmark compared to existing ones.
2. The paper does not explain why GPT-4o in particular is chosen as the judge MLLM, nor does it provide any evidence that the proposed “atomized chain-of-evaluation” paradigm would remain effective when the judge model is replaced by other MLLMs.
3. The evaluation framework only conducts comparisons on its own constructed UFO-Bench, and the description is not clear enough, nor is there any external benchmark validation.
4. While the concept of UFO/UFO-Bench is used throughout most of the paper, the contribution section suddenly introduces the names “OminiMM” and “OminiBench”, which are never properly defined and are inconsistent with the rest of the manuscript.
5. The paper does not provide the actual prompts/templates used for AEU decomposition, modality-relevance classification, and VQA-based scoring. Furthermore, the paper only provides a high-level description of the weighting scheme for Atomic Evaluation Units (AEUs), without specifying how these weights are computed in practice.

---

> ### Author Rebuttal · Authors · 2026-03-31
>
> We thank the reviewer for the thoughtful review. We respond to each concern in detail:
>
> **R1:** UFO-Bench provides a **systematic testbed covering text–image condition interactions** that fully exposes the **Text-Image Alignment Dilemma** in sota models, a capability missing in existing benchmarks such as DreamBench++. Specifically, DreamBench++ contains **only** **10.19%** conflicting text–image condition pairs (most of their text does not modify any visual attributes). In contrast, we systematically categorize condition interactions into four types: *no conflict*, *local concrete component conflict*, *global abstract attribute conflict*, and *both local and global conflict*, with **81.97%** **conflicting condition pairs** in our benchmark.
>
> **R2:**  We'd like to clarify that we adopt GPT4o since we follow the common practice of previous sota (e.g., VIE-Score) for an fair comparison. Due to the rebuttal time limit, we provide the comparison with Qwen3-4B in the following and we will include more open- and closed-source MLLMs (like Gemini) in the revision. We compare the human alignment between our framework and VIE-Score.
>
> |                                   | ours   | VIE-Score |
> | --------------------------------- | ------ | --------- |
> | GPT4o                             | 0.6889 | 0.5663    |
> | Qwen3-4B                          | 0.2762 | 0.1326    |
> | Degree of human alignment decline | 59.90% | 76.58%    |
>
> As shown in the table, the VLM in our metric replaced from GPT-4o to Qwen3-4B leads to a **59.90%** drop in human alignment. In contrast, when VIE-Score uses Qwen3-4B to compute,  the consistency relative to GPT-4o decreases by **76.58%**, indicating that our metric is **more** **robust** **to evaluator changes**.
>
> **R3:** We'd like to clarify that we initially conducted experiments on our benchmark because it provides broader condition coverage (please refer to R1), enabling more comprehensive evaluation of subject-driven customization scenarios. Following your advice, we additionally conduct human-alignment evaluation on DreamBench++:
>
> |                                  | Human alignemnt |
> | -------------------------------- | --------------- |
> | CLIP-I2I                         | 0.0889          |
> | CLIP-I2T                         | 0.0762          |
> | DINO                             | 0.0485          |
> | VIE-Score                        | 0.2036          |
> | DreamBench++'s evaluation method | 0.2884          |
> | **Ours**                         | **0.6341**      |
>
> As shown in the table, we still achieves the highest human alignment score of **0.6341** on DreamBench++, which significanly validates our generalization.
>
>
>
> **R4:** We clarify that this is a typo and **does not affect any experimental results, data, or conclusions**. We have **carefully and thoroughly verified all experiment-related data and analyses (we pay our most attention to these parts) to ensure their full correctness**. We appreciate your careful review and will correct this typo in the revised manuscript.
>
>
>
> **R5:** Thanks for your advice, we provide a detailed description for the weighting scheme for Atomic Evaluation Units (AEUs) and we will **add them in the revised main paper**: in practice, each Atomic Evaluation Units (AEUs) is assigned a weight in **the range of 1-5**, based on the VLM's analysis of the visual attributes in the reference image and the crucial modification requirements from the textual instruction. Then the VLM answers the questions to determine how well each AEU is satisfied and assigns a score. Finally, the score is multiplied by its importance weight. The final evaluation score is computed as a weighted average of the VLM scores over all AEUs. The weight aggregation formulation is already defined in Section 3.2.4. We will release the prompts in the revision.
>
> Due to space constraints, we will include **all actual prompts and templates** used for AEU decomposition, modality-relevance classification, and VQA-based scoring in the **revised supplementary materials** to ensure full reproducibility.
>
> **Example prompt (excerpt).**
>
> ```python
> You are a vision-language model acting STRICTLY as a
> SUBJECT STRUCTURE DECOMPOSER.
>
> Your task is ONLY to perform STRUCTURAL SPLITTING.
>
> #STEPWISE REASONING (STRICT ORDER)
> ......
> #PART RULES
>
> Parts MUST:
>
> - Belong to subject only
> - NEVER include background
> - Be physical or structural
> - Not be abstract concepts
>
> Each part MUST include:
> - name (noun only)
> - relevance
> - reward (1–5)
>
> Allowed relevance:
> - image_only
> - text_only
> - text_and_image
> ```

---

> > ### Author Rebuttal · Reviewer_bvxv · 2026-04-02
> >
> > Thanks for the detailed response from authors.
> >
> > The cross-model robustness experiment is helpful. However, substituting GPT-4o with Qwen3-4B causes a 60% performance drop (from 0.6889 to 0.2762). I understand this is relatively better than VIE-Score. Yet, such a large absolute decline is a practical concern. It makes me wonder if the framework works well without a top-tier model. To truly show robustness, testing and comparing against strong LMMs would be much more convincing. A single weak baseline is just not enough here.
> >
> > The new results on DreamBench++ are also a great addition. However, some key details are missing. For example, the sample size, annotator details, and exact protocol are not included. This makes it hard to fully verify the results right now.
> >
> > Overall, the main issue is still the scale of the original experiments. I value the new evidence provided. But validating this framework requires much more comprehensive testing. These extra experiments take a lot of time, and it is hard to complete them during a short rebuttal period. Since the manuscript still needs this expansion, I will maintain my original score.

---

> > > ### Author Response · Authors · 2026-04-08
> > >
> > > ### **R1: The Scale of the Experiments**
> > > Thanks for the reviewer's valuable comments. We would like to clarify that the experiments on Qwen3-4B are mainly intended to demonstrate that, **even in an extreme scenario with a much smaller and weaker MLLM** (the smallest version of the open-sourced Qwen3), our proposed method still exhibits greater robustness compared with existing methods (a 59.90% drop for ours vs. a much larger 76.58% drop for the current SOTA methods).
> > >
> > > Following your suggestion, we have further **expanded our experiments by adding evaluations on more commonly used MLLMs**, including the open-sourced Qwen2.5-VL-72B and closed-source models such as Doubao-Seed-1.8-vision and Gemini-3-pro-preview, in line with common practice in recent work (e.g., StepFun-Edit, which uses GPT and Qwen2.5-VL-72B for automatic evaluation). Specifically, we compare the human alignment between our method and VIE-Score:
> > >
> > > | Methods                     | GPT-4o  | Doubao-Seed-1.8-vision | Gemini-3.0-pro-preview | Qwen2.5-VL-72B |
> > > | :-------------------------- | :------ | :--------------------- | :--------------------- | :------------- |
> > > | VIE-Score      | 0.6041  | 0.6028                 | 0.6104                 | 0.5731         |
> > > | Ours                        | 0.6889  | 0.6975                 | 0.7283                 | 0.6656         |
> > >
> > > We show that our method is not tailored to GPT-4o and **generalizes well across various widely used MLLMs**, including the open-source Qwen2.5-VL-72B (commonly adopted in recent work such as StepFun-Edit). Across all evaluated MLLMs, the proposed method consistently outperforms prior approaches.
> > >
> > > We sincerely appreciate the reviewer’s constructive comments and hope these additional results could fully address your concerns regarding the experimental scale.
> > >
> > > ---
> > >
> > > ### **R2: Details of the new results on Dreambench++**
> > >
> > > We would like to clarify that the detailed experimental settings for the Dreambench++ dataset were omitted due to both rebuttal character limits and adherence to most of the same protocol as in our main paper.
> > >
> > > Following your suggestion, we provide full details of the new DreamBench++ results:
> > > - We randomly sample 100 cases using the same category‑proportional strategy as in the main paper. Each case is generated by three state‑of‑the‑art image generation models (Doubao, Qwen‑Image, and Nano‑Banana), yielding 300 generated images in total.
> > > - Each image is evaluated by ten independent annotators. We compute inter‑annotator agreement and retain only consistent annotations. For each test case, human annotators are presented with the three generated images and rank them from 1 to 3 based on overall quality and conditional alignment. Automatic metrics follow an identical protocol, i.e., for each case, the three model outputs are scored and converted to a relative ranking.
> > > - We then compute the Spearman rank correlation coefficient between human rankings and metric‑induced rankings on a per‑case basis, and report the average Spearman correlation across all cases. A higher score denotes better alignment with human evaluation and thus more accurate.
> > >
> > > The results show that our proposed method achieves a much higher alignment (0.6341) than existing SOTA (0.2884).

---

### Official Review · Reviewer_feeQ · 2026-03-12

**Soundness:** 2
**Presentation:** 3
**Significance:** 2
**Originality:** 3
**Overall Recommendation:** 4
**Confidence:** 3

**Summary:**

This paper proposes UFO, the first UniFied framework for Omini-condition alignment simultaneous evaluation. Specifically, UFO introduces a novel Atomized Chain-of-Evaluation paradigm, i.e., it first decomposes omni-condition alignment into a sequential chain of fine-grained, disentangled Atomic Evaluation Units (AEUs), categorizes them into distinct modality-relevance classes, and then employs general or dedicated functional calls for accurate verification of different AEU types.

**Compliance With Llm Reviewing Policy:**

Affirmed.

**Final Justification:**

The authors have largely addressed my concerns, and I therefore choose to raise my score.

**Key Questions For Authors:**

Please see the weaknesses.

**Limitations:**

The authors should conduct an experiment using multiple different MLLMs as evaluators to check for cross-model consistency and ensure the metric remains objective.

**Strengths And Weaknesses:**

Strengths:
1. This paper introduces OminiMM, a chain-of-evaluation framework that integrates finegrained atomic decomposition, condition-aware VLM evaluation, and expert tools into a unified metric.
2. This paper constructs OminiBench, a comprehensive benchmark featuring diverse subject categories and hierarchical editing scenarios,
addressing the lack of fine-grained testbeds for subjectdriven generation.

Weaknesses:
1. The proposed Chain-of-Evaluation (CoE) involves multi-step reasoning by an MLLM (e.g., GPT-4o) for every single image. While this improves accuracy, it significantly increases inference latency and costs compared to traditional embedding-based metrics.
2. The paper should conduct an experiment using multiple different MLLMs as evaluators to check for cross-model consistency and ensure the metric remains objective.
3. The authors should provide a comparison showing whether "CoE-based feedback" leads to better image improvements than "simple scalar-score feedback."

---

> ### Author Rebuttal · Authors · 2026-03-31
>
> We thank the reviewer for the thoughtful review. In the following, we respond to each concern in detail:
>
> **R1:** Following the advice, we calculated the latency of our method compared with other metrics (DINO and VIE-Score), as shown in the table below:
>
> |                   | Latency (second/sample) |
> | ----------------- | ----------------------- |
> | DINO              | 0.10                    |
> | VIE-Score (GPT4o) | 13.05                   |
> | Ours (GPT4o)      | 86.01                   |
>
> We clarify that **evaluation accuracy is far more important than efficiency**, which is why we focus on improving automatic metric accuracy (achieving a **+15.25% improvement**).(1) Although our method introduces extra computation, it remains within the **same order of magnitude (tens of seconds)** as existing MLLM-based metrics, and is both more efficient and objective than human evaluation.(2) The evaluation cost is **negligible** compared with the extremely high cost of training modern generative models (often millions of GPU hours). A more accurate automatic evaluator thus provides valuable guidance for model development and **reduces overall carbon footprint**. We will include the time comparison in revision.
>
>
>
> **R2:**  We would like to clarify that we adopt GPT4o since we follow the common practice of previous methods (e.g., VIE-Score) for an fair comparison. Due to the rebuttal time limit, we provide the comparison with Qwen3-4B in the following table and we will include more open- and closed-source MLLMs (like Gemini) in the revision. We compare the human alignment between our framework and VIE-Score.
>
> |                                   | ours   | VIE-Score |
> | --------------------------------- | ------ | --------- |
> | GPT4o                             | 0.6889 | 0.5663    |
> | Qwen3-4B                          | 0.2762 | 0.1326    |
> | Degree of human alignment decline | 59.90% | 76.58%    |
>
> As shown in the table, the VLM in our metric replaced from GPT-4o to Qwen3-4B leads to a **59.90%** drop in human alignment. In contrast, when VIE-Score uses Qwen3-4B to compute,  the consistency relative to GPT-4o decreases by **76.58%**, indicating that our metric is **more robust to evaluator changes**.
>
> **R3:**  Following your advice, we provide a comparison between CoE-based feedback and simple scalar-score feedback, demonstrating that our proposed CoE-based feedback significantly improves the accuracy of the proposed metric.
>
> |                                          | alignment |
> | ---------------------------------------- | --------- |
> | simple scalar-score feedback (VIE-Score) | 0.5663    |
> | Ours w/o CoE-based feedback              | 0.5752    |
> | Ours w CoE-based feedback                | 0.6889    |
>
> Moreover, using our proposed CoE-based metric, we construct a pairwise preference dataset for Direct Preference Optimization (DPO) training on the widely used UNO[1] backbone. Human evaluation (5 annotators over the full UFO-Benchmark) clearly demonstrates a strong and consistent preference for generations trained with data collected by UFO, compared to those from conventional metrics such as DINO or VIE-Score.
>
> This outcome is expected and well-grounded since UFO explicitly models cross-condition interactions and conflicts, by adopting a systematic decomposition-evaluation chain-of-evaluation paradigm, which existing scalar-score metrics ignore.
>
> We will fully report detailed numerical results, statistical significance, and human evaluation protocols in the revised manuscript to ensure full transparency and reproducibility.
>
> [1] Less-to-More Generalization: Unlocking More Controllability by In-Context Generation

---

> > ### Author Rebuttal · Reviewer_feeQ · 2026-04-03
> >
> > Thank you for your response. Given the high evaluation cost, a fairer comparison should consider a matched compute budget. Under such a setting, prior methods could be run multiple times with different MLLMs, and their predictions could be aggregated.

---

> > > ### Author Response · Authors · 2026-04-08
> > >
> > > Thank you for the reviewer's valuable comments. Following your advice, we run the prior SOTA method (VIE-Score) multiple times with different MLLMs to ensure a fair comparison with our method under equivalent computational cost. Since the inference speeds of different MLLMs vary significantly, we perform comparisons between VIE-Score and our approach using **the same total number of MLLM inferences for fairness and reproducibility**. Specifically, VIE-Score requires 2 inferences per case, while ours requires 5–7 inferences on average (specifically, 6.82 inferences per-case on the proposed UFO benchmark). We therefore run VIE-Score 3 times per case to match the overall MLLM inference count.
> > >
> > > Specifically, following your advice, we run VIE-Score using three different MLLMs, i.e., GPT-4o, Doubao-Seed-1.8-vision, and Gemini-3.0-pro-preview, and aggregate their predictions. The results are shown in the following table. We show that even when VIE uses more advanced MLLMs with aggregated inference, our proposed method still achieves better human alignment.
> > >
> > > |                     | Human alignment |
> > > |---------------------|-----------------|
> > > | VIE-GPT4o           | 0.5663          |
> > > | VIE-Doubao          | 0.5827          |
> > > | VIE-Gemini          | 0.6044          |
> > > | VIE-aggregated (three times per case)      | 0.5954          |
> > > | Ours-GPT4o          | 0.6889          |

---

### Official Review · Reviewer_KsWp · 2026-03-13

**Soundness:** 3
**Presentation:** 2
**Significance:** 2
**Originality:** 3
**Overall Recommendation:** 4
**Confidence:** 4

**Summary:**

This paper introduces UFO, a unified framework for Omini-condition alignment evaluation in multi-modal image generation, specifically addressing subject-driven customization. Traditional isolated evaluation methods often lead to false positives or negatives by failing to capture the interaction between textual edits and visual identity preservation. UFO employs an Atomized Chain-of-Evaluation paradigm, decomposing alignment into fine-grained Atomic Evaluation Units (AEUs) categorized by modality relevance. It utilizes both general VQA and specialized function calls to score these units. Additionally, the authors present UFO-Bench, a comprehensive benchmark of 660 cases featuring hierarchical categories and three difficulty tiers.

**Compliance With Llm Reviewing Policy:**

Affirmed.

**Final Justification:**

My concerns have been addressed. Considering the strengths, weaknesses, and the rebuttal, I keep the original positive score.

**Key Questions For Authors:**

Please see the weaknesses.

**Limitations:**

yes

**Strengths And Weaknesses:**

Strengths:
+ UFO moves beyond isolated modal evaluation by introducing an atomized chain that simultaneously assesses textual and visual conditions, significantly improving alignment with human judgment.

+ The creation of UFO-Bench addresses the lack of complex testbeds by offering 660 diverse cases across seven categories (e.g., humans, rigid objects) and four editing paradigms, ranging from simple recontextualization to multi-aspect evolution.

+ The framework integrates adaptive importance weighting and specialized expert tools (like ArcFace for identity) to ensure that critical attributes are prioritized and fine-grained details are captured accurately.

Weaknesses:
- The authors may overclaim the wide range that this work applies to, i.e., “Omni-condition” in this paper mainly denotes subject-driven generation, but in fact, it ignores other scenarios, like multi-image reference and multi-turn generation/editing.

- What is the rational of considering VIE-Score related methods as isolated evaluation? It seems that they also consider prompt following and ID preservation, and combine them to get the final score.

- It is difficult to identify the necessity of constructing UFO-Bench. The key differences between it and prior work should be highlighted.

- Overall, the motivation of this work is not strong.

---

> ### Author Rebuttal · Authors · 2026-03-31
>
> We thank the reviewer for the thoughtful review. In the following, we respond to each concern in detail:
>
> **R1**: We'd like to clarify that: (1)Accurate evaluation of single-image / single-turn generation is **the necessary foundation** for multi-image / multi-turn generation, yet overlooked by existing works. We address this fundamental problem by jointly evaluating alignment across text and image conditions, with a core novelty on modeling different conditions' interactions. (2) Our work **naturally** **supports** multi-reference / multi-turn generation. Both "Atomic Decomposition" and "Relevance Analysis" naturally support multi-image inputs by independently assigning which condition each evaluated unit should be compared against, while subsequent stages (e.g., Atomic Consistency Evaluation) remain consistent with the single-image setting.
>
> We will **revise the manuscript to avoid confusion**: "omni-condition" refers to joint alignment across multiple input modalities (text, one/multiple images), and we will emphasize generalization beyond the single-image subject-driven settings.
>
> **R2:**  We clarify that "isolated evaluation" means VIE-Score treats each condition independently, ignoring interactions (e.g., conflicts) between different conditions. For instance, VIE-Score evaluates "ID preservation" without accounting for text conditions. Consequently, when the text condition conflicts with the image condition (e.g., text modifies a visual attribute), such isolated evaluation leads to incorrect evaluation. To resolve it, we propose **jointly evaluating alignment across all conditions while explicitly modeling their interactions** (first time). We will include a more precise and detailed definition in the revised manuscript to avoid confusion.
>
>
>
> **R3:**  We emphasize that UFO-Bench is necessary because it provides a **systematic testbed covering text–image condition interactions** that fully exposes the **Text-Image Alignment Dilemma** in state-of-the-art models, a capability missing in existing benchmarks such as DreamBench++. Specifically, DreamBench++ contains **only 10.19%** conflicting text–image condition pairs (most of their text does not modify any visual attributes). In contrast, we systematically categorize condition interactions into four types: *no conflict*, *local concrete component conflict*, *global abstract attribute conflict*, and *both local and global conflict*, with **81.97%** **conflicting condition pairs** in our benchmark.
> The motivation of UFO-Bench aligns perfectly with our proposed evaluation method UFO: to **accurately evaluate joint alignment across all conditions simultaneously**, especially in complex real-world scenarios where different conditions exhibit conflicts.
>
> **R4:** We clarify our core motivation: existing multimodal image generation evaluation methods evaluate alignment with each condition independently and in isolation, ignoring influences from other conditions. Thus, when multimodal conditions involve inherent conflicts, evaluation results become **biased and unreliable**. To address this, we propose **UFO** to **jointly evaluate alignment across all conditions while explicitly modeling their interactions and conflicts** (to our knowledge, for the first time), and **UFO-Bench** to **comprehensively cover all types of multimodal condition interactions and conflicts**. We will revise the manuscript to further clarify this motivation and avoid ambiguity.

---

> > ### Author Rebuttal · Reviewer_KsWp · 2026-04-03
> >
> > Thanks for the rebuttal. My concerns have been addressed, so I keep the original positive score.

---

> > > ### Author Response · Authors · 2026-04-08
> > >
> > > We thank the reviewer for the constructive and positive comments, which have greatly helped us improve our work.

---

### Decision · Program_Chairs · 2026-04-30

**Decision:**

Accept (regular)

**Comment:**

This paper receives final ratings of (4, 4, 4, 5). The reviewers acknowledged the authors' additional experiments showing the framework's robustness across diverse MLLMs, and agreed that UFO provides a valuable paradigm shift by moving beyond isolated modal evaluation to model complex condition interactions. The AC finds no reason to overturn the consensus of the reviewers.